# Validation of the Canadian English and French Versions of the Fear of COVID-19 Scale in Quebec Nursing Staff

**DOI:** 10.3390/ijerph22020297

**Published:** 2025-02-17

**Authors:** Céline Gélinas, Marc-André Maheu-Cadotte, Élisabeth Di Nardo, Mélanie Lavoie-Tremblay, Maria Cecilia Gallani, Émilie Gosselin, Christine Maheu, Sylvie D. Lambert, Melissa Richard-Lalonde, Eric Tchouaket Nguemeleu, José Côté

**Affiliations:** 1Ingram School of Nursing, McGill University, Montreal, QC H3A 2M7, Canada; elisabeth.dinardo@mail.mcgill.ca (É.D.N.); christine.maheu@mcgill.ca (C.M.); sylvie.lambert@mcgill.ca (S.D.L.); melissa.richard-lalonde@mail.mcgill.ca (M.R.-L.); 2Centre for Nursing Research and Lady Davis Institute, Jewish General Hospital, Montreal, QC H3T 1E2, Canada; 3Faculty of Nursing, Université de Montréal, Montreal, QC H3T 1A8, Canada; marc-andre.maheu-cadotte@umontreal.ca (M.-A.M.-C.); melanie.lavoie-tremblay@umontreal.ca (M.L.-T.); jose.cote@umontreal.ca (J.C.); 4Faculty of Nursing, Université Laval, Montreal, QC G1V 0A6, Canada; maria-cecilia.gallani@fsi.ulaval.ca; 5School of Nursing, Université de Sherbrooke, Sherbrooke, QC J1K 2R1, Canada; emilie.gosselin3@usherbrooke.ca; 6Cancer Research Program, Research Institute—McGill University Health Centre, Montreal, QC H3H 2R9, Canada; 7Research Centre, St-Mary’s Hospital, Montreal, QC H3T 1M5, Canada; 8Department of Nursing, Université du Québec en Outaouais, Gatineau, QC J7Z 0B7, Canada; eric.tchouaket@uqo.ca; 9CHUM Research Centre, Montreal, QC H2X 0A9, Canada

**Keywords:** psychometrics, instrument validation, predictive validation, ROC curve, coronavirus, SARS-CoV-2, stress disorders, fear, stress, posttraumatic, emotions, nursing

## Abstract

Nursing staff have been at the forefront of the pandemic, reporting high traumatic stress and anxiety levels related to high fear of COVID-19. Recommendations from previous studies include using the Fear of COVID-19 Scale (FCV-19S) as a screening tool to identify any individuals who may benefit from targeted psychological support. Thus far, the accuracy of the Canadian English and French versions of FCV-19S to detect high levels of traumatic stress and anxiety symptoms has not been examined. The objectives of this methodological psychometric study were to examine among nursing staff: (a) the structure and internal consistency of the Canadian versions of the FCV-19S and (b) its ability in detecting high levels of traumatic stress and anxiety symptoms. An anonymous online survey was distributed among nursing staff (*n* = 387) in the province of Quebec (Canada). This survey included the FCV-19S and scales measuring their traumatic stress (PCL-5) and anxiety symptoms (GAD-7). Exploratory factor analysis and receiver operating characteristic (ROC) analyses were performed. The one-factor structure of the FCV-19S was supported (Cronbach alpha = 0.87). The FCV-19S showed better accuracy for the detection of traumatic stress (area under the curve (AUC) 0.75 [95% CI 0.68, 0.82]) in comparison to anxiety symptoms (AUC 0.65 [95% CI 0.60, 0.74]). The FCV-19S may benefit from adaptation for its use in nursing staff and in a future pandemic context.

## 1. Introduction

Since the coronavirus disease 2019 (COVID-19) first emerged in December 2019 and was declared a pandemic three months later [1], all countries have faced significant public health challenges. Registered and licensed practical nurses (hereafter, nursing staff) have been at the forefront of the pandemic and reported high traumatic stress and anxiety levels [2]. Greater risk for nursing staff to contract the virus compared to the general population [3], worries about spreading the virus to their families, shortages in personal protective equipment [4], ever-evolving guidelines [5], lack of proper training [6], and repeatedly witnessing COVID-19 patients dying [7] all contributed to their traumatic stress and anxiety symptoms. Furthermore, the measures implemented by governments to reduce the spread of the virus, such as self-isolation and social distancing, may also have negatively affected the mental health of nursing staff [8,9].

A recent meta-analysis found strong associations between traumatic stress, anxiety, and the fear of COVID-19 [10]. Fear is a psychological response to a real or perceived threat and plays a crucial role in survival by triggering the autonomic nervous system’s [11] “fight or flight” response [12]. Although it is normal to experience some level of fear of infectious diseases such as COVID-19, high and prolonged fear may exert a toll on rational thinking and lead to mental health problems such as traumatic stress and anxiety [13,14]. Traumatic stress can express itself in various forms, ranging from fear- and anxiety-related symptoms to anhedonic, dysphoric, aggressive, or dissociative symptoms [12]. During the pandemic, traumatic experiences such as contracting a severe form of COVID-19, the death of relatives and friends, and repeated exposure to individuals infected with the virus increased, among other hazards, the risk of developing traumatic stress [15,16,17]. Even though anxiety and fear share similar physiological responses, anxiety mainly relates to imprecise or unknown threats or excessive anticipation of future threats [12].

Recommendations from previous studies include using the Fear of COVID-19 Scale (FCV-19S) [14] as a brief screening tool to identify any individuals who may benefit from targeted psychological support, such as counseling and psychotherapy [18]. Previous studies supported the content validity, the single-factor structure, and the internal consistency of the FCV-19S [19]. However, the accuracy of the FCV-19S as a screening tool for traumatic stress and anxiety symptoms requires further validation. This validation could identify an optimal threshold score for the FCV-19S by considering the scale’s sensitivity and specificity levels in detecting traumatic stress and anxiety symptoms and aid in accurately identifying nursing staff in need of targeted psychological support.

To our knowledge, the accuracy of the FCV-19S in reflecting the severity of traumatic stress and anxiety symptoms —specifically, how the score on this scale might also suggest important levels of traumatic stress or anxiety symptoms—has been examined in only a few previous validation studies, and only for the scale’s Arabic, Chinese, Greek, Korean, and Vietnamese versions. Thresholds that might suggest important levels of traumatic stress or anxiety symptoms ranged from 17 and to 19 on the FCV-19S [20,21,22,23,24]. Only one of these studies specifically targeted nursing staff [23]; the others were conducted among the general population [20,22] or with university students [21,24]. Further research focusing on nursing staff is crucial. High fear levels, known to be associated with traumatic stress and anxiety, can potentially compromise clinical decision making [25] and influence the decision to leave their positions or the nursing profession [26,27].

In 2020, the original English version of the scale [14] was adapted into Canadian English and translated into Canadian French using a double forward-backward method [28], and then validated among a sample of 1708 nursing staff in the province of Quebec [29]. A unidimensional scale with a satisfactory internal consistency (Cronbach’s alpha of 0.88 and 0.90 for the Canadian English and French versions, respectively) was found for both versions. However, the ability of the FCV-19S to detect high levels of traumatic stress and anxiety symptoms in nursing staff has not yet been evaluated, particularly in terms of convergent validation, i.e., how well it correlates with related constructs, and criterion validation, i.e., its accuracy in identifying clinically relevant levels of traumatic stress and anxiety symptoms. Although the acute emergency phase of the COVID-19 pandemic has concluded, the ongoing relevance of the FCV-19S is underscored by the potential long-term psychological impacts of the virus [30]. Moreover, continuing to refine this tool can aid in preparedness for future health crises and serve as a model for assessing fear in various public health contexts. Indeed, beyond its relevance during COVID-19, the FCV-19S—or similarly refined scales for fear assessment—can play a pivotal role in guiding public health strategies and workforce preparedness for future pandemics or health crises. Early identification of heightened fear responses enables timely psychological interventions and resource allocation for nursing staff. By establishing clear threshold scores with robust sensitivity and specificity, health systems can more effectively prioritize support services—such as counseling, psychotherapy, and resilience training—for those most at risk.

In Quebec, during the summer of 2021, when this study was conducted, approximately 375,000 people had been infected with COVID-19 since the first reported case in February 2020, and over 11,000 deaths had been attributed to the virus [31,32]. At the time of the study, hospitalization and death rates from COVID-19 were declining [33]. Vaccination campaigns for healthcare workers, including nursing staff, had been in progress for more than six months [32]. By June 2021, about two-thirds of Quebec’s population had received at least one dose of the COVID-19 vaccine, with vaccination coverage among healthcare workers reaching nearly 86% [34]. The government implemented various public health measures starting in early 2020, which began to relax in late May 2021, including the removal of curfews and travel restrictions, the reopening of outdoor dining, and increased allowances for social gatherings in both public and private settings [35].

Thus, the objectives of this study were to examine the structure and internal consistency of the Canadian versions of the FCV-19S, as well as its ability in detecting high levels of traumatic stress and anxiety symptoms.

## 2. Materials and Methods

### 2.1. Design and Sample

A cross-sectional survey design was used for this methodological psychometric study. During the summer of 2021, an anonymous online survey was made available in both Canadian English and French and distributed among nursing staff practicing in the province of Quebec (Canada). Registered nurses’ and licensed practical nurses’ professional orders, the Ordre des infirmières et infirmiers du Québec and the Ordre des infirmières et infirmiers auxiliaires du Québec, collaborated regarding the survey development and distribution. This study (#2021-2451) was approved by the Medical/Biomedical Research Ethics Board of the CIUSSS West-Central Montreal.

Nursing staff were eligible for this study if they had consented to their professional order to be contacted for research purposes at the time of their annual license renewal. There were no additional eligibility criteria. About 28,500 nurses and 24,000 licensed practical nurses had consented in 2021 to be contacted during the study period. To avoid overwhelming nursing staff with research project invitations, 4000 (15%) of those who consented to be contacted were randomly chosen via a probabilistic sampling method and sent an invitation to complete the survey. Based on our previous study using this recruitment method, a participation rate of 10% was expected [29].

### 2.2. Procedures

Randomly selected nursing staff received an email inviting them to complete the online survey. In Quebec (Canada), nursing staff must provide a valid email address when registering with their professional order. This email provided a link leading them to the project invitation letter and an electronic consent form. Once they consented to participate, they had access to the survey hosted on the Qualtrics platform. Nursing participants were informed that the survey was anonymous and could be completed only once. They were also informed that 15 min were estimated for the survey completion. A median duration of 14 min (interquartile range (IQR) from 11 to 20 min) was necessary for nursing participants to complete the survey. The survey was accessible from 9 June 2021 to 31 October 2021.

### 2.3. Instruments

Study data included sociodemographic information and scores from three validated scales: (1) the FCV-19S [14,29]; (2) the shortened (8-item version) Posttraumatic Stress Disorder (PTSD) Checklist for DSM-5 (PCL-5) [36]; (3) the Generalized Anxiety Disorder-7 items (GAD-7) [37,38]. These scales were selected because they aligned with the study objectives; are widely used to assess anxiety and traumatic stress symptoms, including among nursing staff; and were used in previous studies in which the detection accuracy of the FCV-19S was evaluated [20,21,22,23,24,39]. The PCL-5 was expected to correlate with the FCV-19S due to the potential for pandemic-related experiences to induce traumatic stress symptoms [15,16,17], while the GAD-7′s focus on generalized anxiety symptoms aligns with the heightened anxiety levels associated with the excessive anticipation of imprecise or unknown threats [12].

The sociodemographic information included participants’ gender, age, years of experience, whether they worked in a designated COVID-19 healthcare setting, whether they provided care to COVID-19 patients, and whether they had been infected with COVID-19 at work or had a colleague that had been infected. Additionally, participants were asked to evaluate their preparedness to provide care amid the COVID-19 pandemic using a 4-point scale, ranging from 1, for “well prepared”, to 4, for “very poorly prepared”. They were also asked to describe their situation at work during the pandemic, with options from 1, indicating they felt “overwhelmed”, to 4, suggesting they were “not affected”. Preparedness and situation at work scores were dichotomized to ease their interpretation. The sociodemographic questionnaire was prepared and reviewed by the research team and was used in previous studies [29,40,41].

Initially developed for use in the general population, the FCV-19S [14,29] is a 7-item self-reported scale, each rated on a 5-point Likert scale (from strongly disagree (1) to strongly agree (5)). The total score can range from 7 to 35, with a higher score indicating greater fear of COVID-19.

The PCL-5 is an 8-item self-reported scale of traumatic stress, with each item rated on an ordinal scale with five points (from not at all (0) to extremely (5)). The total score can range from 0 to 40, with a higher score suggesting a greater severity of traumatic stress symptoms in the past month. Furthermore, a threshold score of 13 was found to indicate a possible posttraumatic stress disorder and high traumatic stress symptoms [42]. The scale’s internal consistency in this study sample was supported by a Cronbach’s alpha of 0.91.

The GAD-7 [37,38] is a 7-item self-reported scale of anxiety symptoms, each rated on an ordinal scale with four points (from not at all (0) to nearly every day (3)). The total score can range from 0 to 21, with a higher score suggesting a greater severity of the anxiety symptoms over the past two weeks. A threshold score of 10 is considered suggestive of moderate anxiety symptoms [43]. The scale’s internal consistency in this study sample was supported by a Cronbach’s alpha of 0.91.

### 2.4. Data Analysis

Descriptive statistics were used to report the study sample characteristics and scale scores. After identifying identical sociodemographic characteristics responses in open-ended sections, we removed 18 responses from the analysis as potential duplicates. Frequency counts and percentages were used for categorical variables. For continuous variables, data distribution was assessed by considering the asymmetry and kurtosis values, standardized by dividing each by its standard errors, the Kolmogorov–Smirnov and the Shapiro–Wilk tests, and by inspecting the histograms and the Q–Q plots. The medians and the 25th and 75th percentiles (quartiles) were described for each variable. Means and standard deviations (SD) of the FCV-19S were also reported.

An exploratory factor analysis was performed to evaluate the internal structure of the FCV-19S. Assumptions of exploratory factor analysis were assessed using the correlation matrix, the Kaiser–Meyer–Olkin measure, and the Bartlett’s test of sphericity. For factor selection, we used criteria based on both the eigenvalues and the scree plot test, retaining factors with eigenvalues greater than one and that appeared before the point of inflection on the scree plot [44]. For all analyses, an alpha of ≤0.05 was set as the level of significance. Cronbach’s alpha coefficients of the FCV-19S were also reported. Cronbach’s alpha coefficient values > 0.7 were indicative of satisfactory internal consistency [45].

For convergent validation, associations between the FCV-19S, the GAD-7, and the PCL-5 scores were evaluated using Spearman’s correlation coefficients. The correlation coefficients were interpreted as follows: 0 to 0.39, weak; 0.4 to 0.6, moderate; >0.7, strong [46]. We hypothesized that the scores for the FCV-19S would be positively correlated with those for the GAD-7 and PCL-5. The Kruskal–Wallis H and Mann–Whitney U tests were also used to explore differences in participants’ FCV-19S scores based on sociodemographic characteristics (described in Section 2.3). We hypothesized that higher scores on the FCV-19S would be associated with fewer years of experience (both in the profession and in their current healthcare establishment), perceptions of not feeling prepared, and a sense of being overwhelmed in the workplace. Lastly, we hypothesized that participants with more direct exposure to COVID-19—such as personal infection, infection of colleagues, caring for COVID-19 patients, or caring for COVID-19 patients who subsequently died—would result in higher scores on the FCV-19S.

For criterion validation, receiver operating characteristic (ROC) analyses were performed to explore the accuracy of the FCV-19S in detecting high traumatic stress (score ≥ 13 for the PCL-5) and moderate anxiety symptoms (score ≥ 10 for the GAD-7). Sensitivity and specificity were also reported for a selection of FCV-19S thresholds. The highest Youden’s index value was used to select the most optimal threshold on the FCV-19S to maximize sensitivity and specificity. The area under the curve (AUC) obtained by the ROC analyses was interpreted as follows: 0.5 to 0.69, low accuracy; 0.7 to 0.89, moderate accuracy; 0.90 and over, high accuracy [47].

## 3. Results

### 3.1. Description of Study Sample and Scale Scores

A total of 387 study participants were included in the analyses. Participants had a mean age of 42 (SD = 12) and were mostly women (*n* = 341, 88.1%). Participants had a mean of 17 years of experience in their role (SD = 11.90) and 10 years (SD = 9.46) in their healthcare setting. Few participants (*n* = 31, 8.0%) identified as licensed practical nurses. More than half of nursing staff worked in a designated COVID-19 setting (*n* = 217, 56.1%) and provided care to COVID-19 patients (*n* = 212, 54.8%). For two-thirds of this subgroup of participants, at least one patient to whom they provided care died. Most participants reported that they had a colleague who had been infected with COVID-19 at work (*n =* 254, 65.6%), although a minority reported having been infected themselves with COVID-19 at work (*n* = 48, 12.4%). Most participants reported feeling prepared or well prepared to provide care during the COVID-19 pandemic (*n =* 309, 76.5%) and feeling in control or unaffected by the COVID-19 pandemic at work (*n* = 225, 55.8%).

On the PCL-5, participants had a median score of 4.0 (IQR = 1.0–9.0), with 69 (17.8%) who obtained a score (≥13) indicative of high traumatic stress symptoms. Also, participants had a median score of 4.0 (IQR = 1.0–7.0) on the GAD-7, with 59 (15.2%) who obtained a score (≥10) indicative of moderate anxiety symptom levels.

Item characteristics and the total score on the FCV-19S are presented in Table 1. Individual item scores were mostly positively skewed (skewness > 0), with participants mostly selecting the lowest score options. Kurtosis values varied, with some item scores characterized by heavier tails and sharper peaks (leptokurtic, kurtosis > 0), while others exhibited lighter tails and flatter peaks (platykurtic, kurtosis < 0). The total score distribution was positively skewed. Results for the Canadian English and French versions of the FCV-19S are presented in Appendix A.

Over half of the participants strongly disagreed with three of the scale’s seven items (item 3, *n* = 201, 51.9%; item 6, *n* = 229, 59.2%; item 7, *n* = 209, 54.0%), leading to an overall lower score on the FCV-19S. A total of 34 participants (8.9%) obtained the lowest scale score of 7, and the highest scale score, reported by one participant (0.3%), was 33. Data distribution for the FCV-19S is presented in Appendix B.

### 3.2. Structural Validation

The assumptions were fulfilled in order to perform an exploratory factor analysis. All seven items were intercorrelated and exhibited correlation coefficients ranging from 0.38 to 0.77 (see Appendix C). The overall Kaiser–Meyer–Olkin measure was 0.85, with individual item values ranging from 0.80 to 0.91, supporting the sampling adequacy. Bartlett’s test of sphericity was statistically significant (*p* < 0.001) supporting that the correlation matrix between items was different from zero.

A single-factor structure was supported with an eigenvalue >4, explaining 50.3% of the total variance of the FCV-19S scale (see Table 2). The visual inspection of the scree plot also supported the single-factor structure (see Appendix D). Overall, the FCV-19S total scale displayed a satisfactory Cronbach’s alpha of 0.87 (*n* = 383).

### 3.3. Convergent Validation

The FCV-19S was positively and weakly correlated with both the PCL-5 (Spearman’s r = 0.39) and the GAD-7 (Spearman’s r = 0.30) total scores (both *p* < 0.001). Lower FCV-19S scores were found in participants who self-reported being well-prepared to provide care during the COVID-19 pandemic (median = 14.0 [10.0–18.0]) vs. those who reported being poorly prepared (median of 15 [12.0–19.0]), (*U* = 10 748.5, *z* = −2.41, *p* = 0.016). Lower FCV-19S scores were also found in regards to participants feeling in control of the situation at work (median of 13.0 [10.0–18.0] vs. those who felt overwhelmed (median of 15.0 [11.0–19.0]), (*U* = 15 344.5, *z* = −2.38, *p* = 0.018).

However, years of experience in the nursing profession (Spearman’s r = 0.03, *p* = 0.95) or in their healthcare setting (Spearman’s r= −0.03, *p* = 0.52) were not associated with FCV-19S total scores. Regarding COVID-19 exposure, FCV-19S total scores were not associated with having been infected with COVID-19 at work (*U* = 6 790.5, *z* = −1.72, *p* = 0.09), having had a colleague infected at work (*U* = 16 012.5, *z* = −0.23, *p* = 0.82), having cared for a COVID-19 patient (*U* = 17 080, *z* = −1.01, *p* = 0.31), nor having cared for a COVID-19 patient who died (*U* = 4 826.5, *z* = −0.26, *p* = 0.80).

### 3.4. Criterion Validation

ROC curves of the FCV-19S regarding traumatic stress and anxiety symptoms are presented in Figure 1 and Figure 2, and a selection of thresholds that maximized sensitivity and specificity levels are described in Table 3.

Using the PCL-5 threshold score of 13, a moderate level of accuracy of the FCV-19S (AUC 0.75 [95%CI 0.68, 0.82], *p* < 0.001) was found for the detection of high traumatic stress symptoms. Using Youden’s index, an FCV-19S threshold score of 17 could maximize sensitivity and specificity.

Using the GAD-7 threshold score of 10, a low level of accuracy of the FCV-19S was found (AUC 0.65 [95%CI 0.57, 0.74], *p* < 0.001) for the detection of moderate levels of anxiety symptoms. Using Youden’s index, an FCV-19S threshold score of 15 was optimal to maximize sensitivity and specificity.

## 4. Discussion

In this sample of 387 nursing staff from the province of Quebec (Canada), and similar to the results of our initial validation study [29], the FCV-19S showed a single-factor structure and satisfactory internal consistency. The scale was associated with stress symptoms and demonstrated moderate accuracy to detect high traumatic stress and low accuracy to detect moderate anxiety symptoms. Nursing staff who self-reported being well prepared to provide care and feeling in control of the situation at work reported lower FCV-19S scores than those who did not. These study findings advance the field by reporting ROC analyses, which have been scarcely used in previous studies. In addition, the validation of the FCV-19S in nursing staff was needed, as this group has been underrepresented in previous validation work.

The FCV-19S score showed a better accuracy for the detection of traumatic stress compared to anxiety symptoms, as suggested by both our ROC analyses and the convergent validation results. This may stem from the conceptual relationship between fear and traumatic stress, which may differ from anxiety. Traumatic stress involves direct or indirect exposure to a traumatic event, such as death or a severe threat to one’s life (for example, the direct threat that COVID-19 poses to nursing staff). However, anxiety is not necessarily linked to a specific traumatic event but may arise from various factors, such as genetics and personality [12]. In this context, both the FCV-19S and the PCL-5 focus on nursing staff reactions or responses to stressful or fear-inducing events, while the GAD-7 emphasizes the general mental state of nursing staff in recent weeks. Furthermore, the high traumatic stress related to COVID-19 among nursing staff might lead to numbness or desensitization to other anxiety-related stimuli. This could result in a narrowed focus on the traumatic stress-inducing event or even overshadow and mask underlying anxiety [48,49]. While previous research [20,21,22,23,24] primarily emphasized anxiety and depressive symptoms, additional studies are needed to elucidate how the FCV-19S might correlate with traumatic stress.

Furthermore, using Youden’s index—a well-recognized method for selecting optimal thresholds that balance sensitivity and specificity in healthcare [50]—we found that a threshold score of 17 was optimal for detecting high traumatic stress, while a score of 15 was optimal for moderate anxiety. Previous research teams who also used Youden’s index to determine threshold scores for the FCV-19S in relation to anxiety or traumatic stress suggested slightly higher thresholds—namely, 19 for traumatic stress and 17 to 19 for anxiety [20,21,22,23,24]. However, as previously reported, the primary goal of identifying an optimal threshold was predictive, i.e., for identifying individuals who might benefit most from further evaluation or intervention, rather than diagnostic of mental health conditions. In the present study, we described various threshold scores of the FCV-19S, allowing researchers and clinicians to choose the most appropriate threshold based on their specific needs, prioritizing either higher sensitivity or specificity.

Consistent with previous research [19], our findings supported the single-factor structure and internal consistency of the FCV-19S. However, most of these previous studies focused on the general population and were predominantly conducted in Asia [51,52]. To our knowledge, only one study led by our research team [29] was conducted among nursing staff in Canada, and its findings are consistent with those obtained in this study. Furthermore, another study conducted among a general population in Quebec (Canada) also found a unidimensional factor structure with a satisfactory internal consistency [53]. These updated findings in a Canadian English- or French-speaking nursing staff sample underscore the scale’s continued reliability over time and across diverse contexts.

Our findings suggest that nursing staff who perceived themselves as well-prepared and in control at work reported lower fear of COVID-19. This relationship may be attributed to greater organizational support and increased confidence in their ability to manage challenges and protect themselves and others. The critical role of organizational support in enhancing nursing staff’s psychological well-being during the COVID-19 pandemic has been highlighted in prior research [54,55]. For instance, access to adequate resources, clear guidelines, and well-defined protocols could contribute to a heightened sense of control, thereby alleviating fear. However, characteristics such as years of experience, work context, and even direct exposure to COVID-19 were not significantly associated with fear levels in this study. Reviews that have explored potential associations between sociodemographic characteristics and levels of fear of COVID-19 have found mitigated and heterogeneous results [51,52]. These study results suggest the importance of preparedness and perceived control in mitigating fear. These findings also suggest that the development of targeted training and support programs focusing on nursing staff’s preparedness and sense of control might be beneficial in alleviating fear.

The total FCV-19S scores from our study participants were notably low, consistent with findings from a meta-analysis of 10 studies, mainly conducted in Asia and Europe, which reported a mean score of 13.11 among healthcare professionals [52]. Furthermore, the total FCV-19S scores were slightly lower than those obtained in our initial study among the same population about a year prior to this study [29]. Several underlying factors might account for these observations. First, despite our survey’s anonymity and online format, participants could have hesitated to reveal their genuine fears related to COVID-19. Secondly, the availability and adoption of the COVID-19 vaccine prior to the commencement of the study in early 2021 might have substantially mitigated their fear levels [56]. Lastly, there is the potential impact of institutional protective measures. Prior research emphasized that implementing protective measures against COVID-19 in clinical settings may have contributed to decreasing healthcare professionals’ concerns [57]. In Quebec, initial personal protective equipment shortages during the early pandemic [58] were offset by enhanced infection control protocols [59], possibly reducing fear among nursing staff. Consequently, nursing staff might have felt adequately prepared and mostly unimpacted by COVID-19 challenges in their healthcare setting during this period (summer and early fall of 2021).

Still, the relatively low total FCV-19S scores might hint at issues when assessing their level of fear. This could create potential challenges in detecting variations, especially among those with milder levels of fear, and in distinguishing significant shifts at the scale’s lower spectrum. Consequently, this may reduce the scale’s utility and validity in tracking differences over time or post-intervention [60]. Further refinements of the scale could be considered to better capture the full spectrum of fear. With a larger sample, an item response theory (IRT) analysis could evaluate the discrimination and difficulty of individual items within the FCV-19S, ensuring their proper functioning across varying levels of fear assessment. Additionally, increasing the number of anchor points on the response scale and refining the wording of response options may enhance the scale’s sensitivity to subtle variations in fear levels. Furthermore, given that the FCV-19S was initially developed for the general public and not specifically for healthcare workers, such as nursing staff, adapting the scale for this specific population should be considered. Indeed, the scale may not fully capture the unique fears and stressors experienced by healthcare workers, potentially influencing their self-assessment. Adapting the scale content to reflect the context of nursing staff could enhance its validity for this population. Nursing staff face unique fears that may differ from those of the general public, such as fear of infecting loved ones due to workplace exposure, concerns about personal safety and availability of personal protective equipment, and fear of stigmatization [3,4,5,6,7]. Due to their training and experience, they may also perceive and respond to fear differently. Incorporating findings from literature reviews and focus groups on the unique fears and stressors of nursing staff could provide qualitative data to guide the adaptation of the FCV-19S for this specific population.

This study has several strengths and limitations that warrant consideration. Among its strengths is its recruitment strategy, soliciting nursing staff from various healthcare settings across the province of Quebec (Canada) via a probabilistic sampling method. This study sample closely mirrors the target population regarding key sociodemographic characteristics in 2021 when the study was conducted, such as age (41.1 years old among nursing staff in Quebec, compared to 42 years old in our sample) and gender (88.4% identifying as women among nursing staff in Quebec, compared to 88.1% in our sample), enhancing the external validity of our findings [61]. Moreover, selecting standardized and validated scales ensured the rigorous measurement of traumatic stress and anxiety symptoms [36,37,38]. Additionally, our investigation provided an updated perspective on nursing staff mental health after the implementation of the vaccine. However, as the study was cross-sectional, it is impossible to infer from the data whether participants’ fear of COVID-19 preceded their traumatic stress or anxiety symptoms. Additionally, we did not include covariates in our convergent validation analyses, as this was judged to fall outside of the scope of this work. Still, studies that aim to identify factors related to the fear of COVID-19, such as workplace characteristics, should consider multivariate models to provide a more comprehensive and accurate understanding. Furthermore, most nursing staff filled out the Canadian French version of the FCV-19S, which led to a relatively low number of nursing staff who used the Canadian English version. However, the psychometric properties and response patterns for the FCV-19S were consistent for both versions and with our initial validation findings [29]. Finally, further psychometric properties of the FCV-19S should be explored, such as measurement error, test–retest reliability, and responsiveness. Longitudinal and experimental research designs would allow for the evaluation of changes in COVID-19 fear levels over time, as well as the evaluation of scale responsiveness to intervention effects.

## 5. Conclusions

The FCV-19S showed a single-factor structure and satisfactory internal consistency in this nursing staff sample. Furthermore, the FCV-19S demonstrated moderate accuracy in identifying traumatic stress and low accuracy in identifying anxiety. Additional studies are needed to elucidate how the FCV-19S might correlate with traumatic stress and explore other psychometric properties. Preparedness to provide safe care in a pandemic context and a sense of control in the work setting were related to lower fear levels in nursing staff and deserve attention in the future development of support interventions. As nursing staff face unique stressors and may perceive and respond to fear differently, future studies should focus on adapting the FCV-19S for this population and in a future pandemic context.

## Figures and Tables

**Figure 1 ijerph-22-00297-f001:**
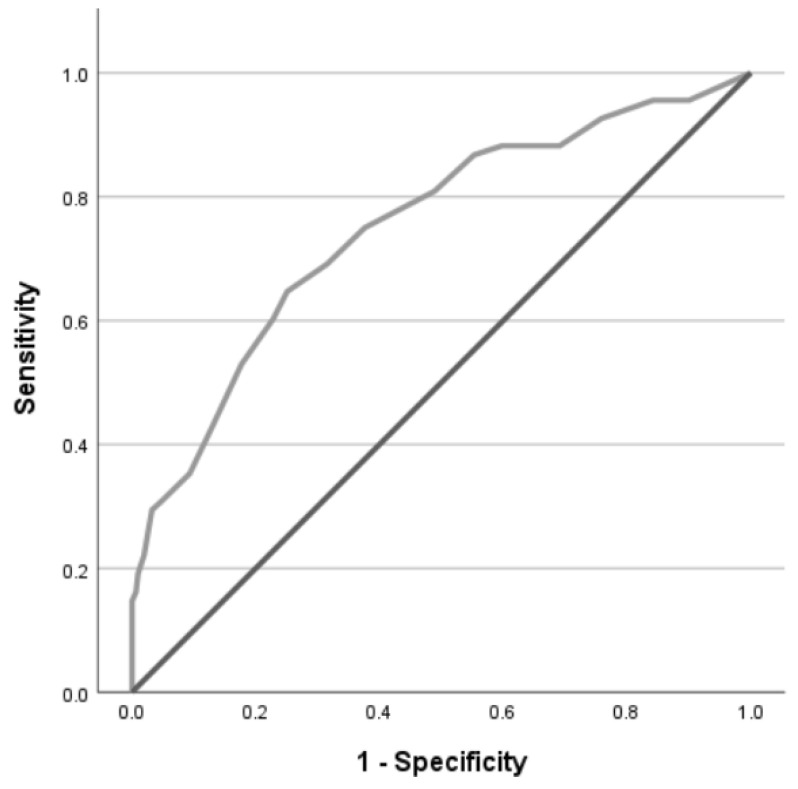
Receiver operating characteristic (ROC) curve of the FCV-19S for the detection of high traumatic stress symptoms (PCL-5 ≥ 13) among nursing staff (*n* = 383).

**Figure 2 ijerph-22-00297-f002:**
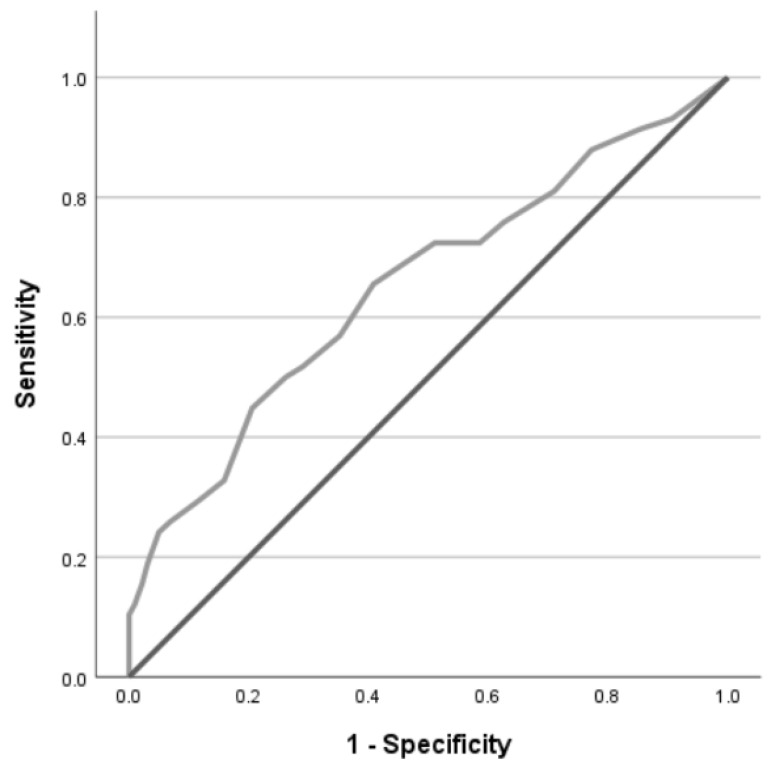
Receiver operating characteristic (ROC) curve of the FCV-19S for the detection of a moderate level of anxiety symptoms (GAD-7 ≥ 10) among nursing staff (*n* = 383).

**Table 1 ijerph-22-00297-t001:** Item characteristics and the total scores of the Canadian English and French versions [29] of the Fear of COVID-19 Scale (FCV-19S).

Items	*n*	Mean (SD)	Median (Q25–Q75)	Skewness (SE)	Kurtosis (SE)
1. I am very afraid of COVID-19	387	2.57 (1.10)	2.00 (2.00–3.00)	0.39 (0.12)	−0.60 (0.25)
2. It makes me uncomfortable to think about COVID-19	385	2.56 (1.17)	2.00 (2.00–4.00)	0.19 (0.12)	−1.03 (0.25)
3. My hands become clammy when I think about COVID-19	385	1.64 (0.81)	1.00 (1.00–2.00)	1.30 (0.12)	1.65 (0.25)
4. I am afraid of losing my life because of COVID-19	385	1.97 (1.11)	2.00 (1.00–3.00)	1.06 (0.12)	0.34 (0.25)
5. I become nervous or anxious when watching news and stories about COVID-19 on social media	386	2.52 (1.19)	2.00 (1.00–4.00)	0.20 (0.12)	−1.16 (0.25)
6. I cannot sleep because I’m worrying about getting COVID-19	386	1.57 (0.81)	1.00 (1.00–2.00)	1.50 (0.12)	2.08 (0.25)
7. My heart races or palpitates when I think about getting COVID-19	385	1.71 (0.94)	1.00 (1.00–2.00)	1.30 (0.12)	1.08 (0.25)
Total score	383	14.55 (5.42)	14.00 (10.00–18.00)	0.64 (0.13)	0.04 (0.25)

Note. All items are scored from 1 (strongly disagree) to 5 (strongly agree). The total score can range from 7 to 35, with higher scores indicating greater fear of COVID-19. SD: standard deviation; Q25: quartile 25%; Q75: quartile 75%; SE: standard error.

**Table 2 ijerph-22-00297-t002:** Exploratory factor analysis of the Canadian versions of the Fear of COVID-19 Scale (FCV-19S) (*n* = 383).

Factor	Initial Eigenvalues			Extraction Sums of Squared Loadings		
	Total	% of Variance	Cumul. %	Total	% of Variance	Cumul. %
1	4.02	57.5	57.5	3.52	50.3	50.3
2	0.83	11.8	69.2			
3	0.64	9.1	78.3			
4	0.53	7.6	86.0			
5	0.41	5.9	91.8			
6	0.31	4.5	96.3			
7	0.26	3.7	100.0			

**Table 3 ijerph-22-00297-t003:** Selection of thresholds that maximized sensitivity and specificity levels of the FCV-19S for the detection of high traumatic stress and moderate anxiety symptoms (*n* = 383).

	To Detect High Traumatic Stress ^1^			To Detect Moderate Anxiety ^2^		
Threshold	Sensitivity	Specificity	Youden’s Index	Sensitivity	Specificity	Youden’s Index
10	92.6%	24.1%	0.17	87.9%	22.7%	0.11
11	88.2%	30.9%	0.19	81.0%	29.0%	0.10
12	88.2%	40.2%	0.28	75.9%	37.4%	0.13
13	86.8%	44.7%	0.32	72.4%	41.4%	0.14
14	80.9%	51.1%	0.32	72.4%	48.9%	0.21
15	75.0%	62.4%	0.37	65.5%	59.2%	0.25
16	69.1%	68.5%	0.38	56.9%	64.8%	0.22
17	64.7%	74.9%	0.40	51.7%	71.0%	0.23
18	60.3%	77.2%	0.38	50.0%	73.8%	0.24

Note. ^1^ High traumatic stress is defined as a threshold score ≥ 13 on the short form of the PTSD Checklist for DSM-5 (PCL-5). ^2^ A moderate level of anxiety is defined as a threshold score ≥ 10 on the Generalized Anxiety Disorder-7 scale (GAD-7).

## Data Availability

The data presented in this study are available on request from the corresponding author. However, these data are not openly accessible due to the conditions stipulated in the ethical approval regarding privacy and confidentiality.

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
