# Peer review of "Validation of the Canadian English and French Versions of the Fear of COVID-19 Scale in Quebec Nursing Staff"

_ijerph, 2025, doi:10.3390/ijerph22020297_

Round 1
Reviewer 1 Report
Comments and Suggestions for Authors
I thank the authors for their important work. The paper is very interesting and contributes to the availability of a useful tool with obvious clinical impact. The literature is well summarized, fluent and argued. I only request these two additions:
1) Perhaps elaborating with a few lines on the impact of Covid-19 on your population (e.g., number of deaths, impact on the economy and labor, etc.) would be helpful.
2) Briefly describe the measures taken by your government during the pandemic and subsequent phases.
3) more discussion of sociodemographic risk factors.
The rest of the paper seems to me to be very high in quality. Excellent description of the procedure of the analyses, complete and detailed discussion. Probably, more space could be devoted to practical implications and future research. The sample is adequate. I noticed that the test-retest was not conducted: this should be discussed.
Author Response
Comment 1: I thank the authors for their important work. The paper is very interesting and contributes to the availability of a useful tool with obvious clinical impact. The literature is well summarized, fluent and argued. I only request these two additions:
Response 1: We want to thank the reviewer for their encouraging feedback. We are pleased that the paper was found interesting and that the tool's clinical impact was evident. The recognition of the literature summary is greatly appreciated.
Comment 2: 1) Perhaps elaborating with a few lines on the impact of Covid-19 on your population (e.g., number of deaths, impact on the economy and labor, etc.) would be helpful.
Response 2: In response to the reviewer's comment, we've included a new paragraph towards the end of the introduction that highlights the impact of COVID-19 on our population during the study period. (L103-106)
Comment 3: 2) Briefly describe the measures taken by your government during the pandemic and subsequent phases.
Response 3: In response to the reviewer's comment, we have added a sentence detailing the measures implemented by the government since the onset of the pandemic in early 2020, and how these measures began to relax in late May 2021 (L106-113).
Comment 4: 3) more discussion of sociodemographic risk factors.
Response 4: Following the reviewer's comment, we have added a paragraph in the discussion section that elaborates on the association between participants' characteristics and their scores on the Fear of COVID-19 Scale. (L360-374)
Comment 5: The rest of the paper seems to me to be very high in quality. Excellent description of the procedure of the analyses, complete and detailed discussion. Probably, more space could be devoted to practical implications and future research. The sample is adequate. I noticed that the test-retest was not conducted: this should be discussed.
Response 5: Thank you for your positive feedback on the quality of our paper. We appreciate your suggestions and have addressed them accordingly. In response to the reviewer's comment, we have further elaborated on the practical implications of our results in the discussion (L371-374), and we've also expanded on potential directions for future research, including the need to examine test-retest reliability. (L406-414; L431-434)
Reviewer 2 Report
Comments and Suggestions for Authors
Manuscript ID: ijerph-3377688
This is an interesting study that examines the structure and internal consistency of the Canadian English and French versions of the Fear of Covid-19 Scale (FCV-19S), and its discriminative ability to detect levels of traumatic stress and anxiety symptoms among nursing staff. The work methodology is clear. The techniques used for data collection and processing also meet the scientific criterion.
To improve the manuscript, only two minor modifications may be considered.
Adapt the Keywords to the text of the abstract. Moreover, some words (e.g. coronavirus, SARS-CoV2, Stress Disorders, etc.) are not included in the text of the abstract.
The conclusion should not contain self-citations, even when it concerns the article from which this research was derived. This can be emphasized in other parts of the manuscript.
Author Response
Comment 1: This is an interesting study that examines the structure and internal consistency of the Canadian English and French versions of the Fear of Covid-19 Scale (FCV-19S), and its discriminative ability to detect levels of traumatic stress and anxiety symptoms among nursing staff. The work methodology is clear. The techniques used for data collection and processing also meet the scientific criterion. To improve the manuscript, only two minor modifications may be considered.
Response 1: We want to thank the reviewer for their feedback and hope we were able to address each one of their comments.
Comment 2: Adapt the Keywords to the text of the abstract. Moreover, some words (e.g. coronavirus, SARS-CoV2, Stress Disorders, etc.) are not included in the text of the abstract.
Response 2: We thank the reviewer for this valuable suggestion. While we understand the importance of aligning keywords with the text of the abstract, as recommended in keyword selection guides (https://doi.org/10.1098/rspb.2024.1222), our intention is to have the keywords complement rather than duplicate the terms found in the title and abstract, in order to enhance discoverability. However, if the reviewer considers any of the selected keywords to be irrelevant to the presented article, we are open to modifying them accordingly. Furthermore, if the journal editors prefer the keywords to repeat exact terms from the title or abstract, we are also open to making those adjustments. Still, we have made minor modifications to the keywords to remove synonyms of COVID-19. (L34-36)
Comment 3: The conclusion should not contain self-citations, even when it concerns the article from which this research was derived. This can be emphasized in other parts of the manuscript.
Response 3: Following revisions, the self-citation section has been removed from the conclusion. (L436)
Reviewer 3 Report
Comments and Suggestions for Authors
Dear Authors,
I would like to appreciate and happy to read your interesting works.However, when examining the Fear of COVID-19 scale developed and validated among English and other speaking populations, I noticed several quality issues in the paper.
In comparison to other works such as those by (Ahorsu et al., 2022)(Ullah et al., 2022) (Blázquez-Rincón et al., 2022)(Munandar et al., 2022)(Lin et al., 2021) (Sawicki et.al, 2022), it is crucial for the authors to address not only contextual gaps but also to introduce novel conceptual and methodological gaps. By doing so, the authors can enhance the robustness and significance of their research, ensuring that it stands out in the field.
References
Sawicki, A. J., Å»emojtel-Piotrowska, M., Balcerowska, J. M., Sawicka, M. J., Piotrowski, J., Sedikides, C., Jonason, P. K., Maltby, J., Adamovic, M., Agada, A. M. D., Ahmed, O., Al-Shawaf, L., Appiah, S. C. Y., Ardi, R., Babakr, Z. H., Bălţătescu, S., Bonato, M., Cowden, R. G., Chobthamkit, P., . . . Zand, S. (2022). The fear of COVID-19 scale: Its structure and measurement invariance across 48 countries. Psychological Assessment, 34(3), 294–310. https://doi.org/10.1037/pas0001102
The Ahorsu, D. K., Lin, C. Y., Imani, V., Saffari, M., Griffiths, M. D., & Pakpour, A. H. (2022). The Fear of COVID-19 Scale: Development and Initial Validation. International Journal of Mental Health and Addiction, 20(3), 1537–1545. https://doi.org/10.10. International Journal of Mental Health and Addiction, 20(3), 1537–1545.
Blázquez-Rincón, D., Durán, J. I., & Botella, J. (2022). The Fear of COVID-19 Scale: A Reliability Generalization Meta-Analysis. Assessment, 29(5), 940–948. https://doi.org/10.1177/1073191121994164
Lin, C. Y., Hou, W. L., Mamun, M. A., Aparecido da Silva, J., Broche-Pérez, Y., Ullah, I., Masuyama, A., Wakashima, K., Mailliez, M., Carre, A., Chen, Y. P., Chang, K. C., Kuo, Y. J., Soraci, P., Scarf, D., Broström, A., Griffiths, M. D., & Pakpour, A. H. (2021). Fear of COVID-19 Scale (FCV-19S) across countries: Measurement invariance issues. Nursing Open, 8(4), 1892–1908. https://doi.org/10.1002/nop2.855
Munandar, D., Narimawati, U., Khan, N., Mohsin, M., & Sailan, M. (2022). Fear of Covid-19 Scale: International Journal of Educational Administration, Management, and Leadership, 33–40. https://doi.org/10.51629/ijeamal.v3i1.88
Ullah, I., Jaguga, F., Ransing, R., Pereira-Sanchez, V., Orsolini, L., Ori, D., de Filippis, R., Pakpour, A. H., Adiukwu, F., Kilic, O., Hayatudeen, N., Shoib, S., Ojeahere, M. I., Nagendrappa, S., Handuleh, J. I. M., Dashi, E., Musami, U. B., Vahdani, B., Ashrafi, A., … Ramalho, R. (2022). Fear During COVID-19 pandemic: Fear of COVID-19 Scale Measurement Properties. International Journal of Mental Health and Addiction, 20(4), 2493–2502. https://doi.org/10.1007/s11469-021-00528-9
Author Response
Comment 1: Dear Authors, I would like to appreciate and happy to read your interesting works. However, when examining the Fear of COVID-19 scale developed and validated among English and other speaking populations, I noticed several quality issues in the paper.
Response 1: We want to thank the reviewer for their feedback. We hope our response was able to address their concerns.
Comment 2: In comparison to other works such as those by (Ahorsu et al., 2022)(Ullah et al., 2022) (Blázquez-Rincón et al., 2022)(Munandar et al., 2022)(Lin et al., 2021) (Sawicki et.al, 2022), it is crucial for the authors to address not only contextual gaps but also to introduce novel conceptual and methodological gaps. By doing so, the authors can enhance the robustness and significance of their research, ensuring that it stands out in the field. [followed by many references]
Response 2: We appreciate the reviewer's feedback on addressing conceptual and methodological gaps. In our study, we have extended the validation of the Fear of COVID-19 Scale to a new context by focusing specifically on nursing staff, a group that has been underrepresented in previous research, which predominantly focused on general populations. Additionally, we introduced a novel methodological approach by assessing the criterion validity of the scale, an aspect that has been rarely explored. This allowed us to evaluate the scale's relevance in indicating greater psychological distress through fear, anxiety, and post-traumatic stress symptoms within this professional group, thus contributing both methodologically and conceptually to the literature. Following the reviewer's comment, we have highlighted these contributions in the beginning of the discussion section. (L321-324)
Reviewer 4 Report
Comments and Suggestions for Authors
Dear Editor,
Thank you for the opportunity to review the manuscript titled "Validation of the Canadian English and French versions of the Fear of COVID-19 Scale in Quebec Nursing Staff." Below, I present several suggestions that I believe could enhance the manuscript and improve its academic impact:
Study Design
The use of a cross-sectional design is appropriate for psychometric studies aimed at validating instruments in a specific context.
Including a diverse sample of nursing staff from Quebec enhances the representativeness and generalizability of the results within this group.
Sampling and Sample Size, The probabilistic method for participant selection strengthens external validity.
Although the expected participation rate (10%) is low, it is typical for online surveys and was handled appropriately.
Instruments, The selection of validated scales, such as the FCV-19S, PCL-5, and GAD-7, ensures the quality of the measurements.
The use of both English and French versions reflects adequate cultural and linguistic sensitivity to the Canadian context.
Statistical Analysis, The combination of internal consistency analysis (Cronbach's alpha), structural validation (exploratory factor analysis), and ROC analysis demonstrates a comprehensive approach to evaluating psychometric properties.
The interpretation of sensitivity, specificity, and AUC (area under the curve) values is appropriate.
Areas for Improvement
While a probabilistic method is mentioned, it is not entirely clear how potential selection or non-response biases were addressed. Including an analysis comparing participants and the target population would strengthen the manuscript.
Instruments, The article mentions that the FCV-19S was developed for the general population rather than nursing staff. It would be helpful to further discuss the implications of this limitation and how the instrument could be adapted for healthcare professionals.
Study Duration, Although the data collection period (June to October 2021) is reasonable, there is no detail on how contextual changes (e.g., health policy updates or vaccine availability) were managed, which may have influenced participant responses.
Contextual Variables, Expanding on how workplace characteristics (e.g., perceived stress levels or workload during the pandemic) were controlled or analyzed as covariates would add depth to the analysis.
Threshold Justification
The selection of thresholds for interpreting the FCV-19S's sensitivity and specificity could benefit from a more detailed theoretical or empirical discussion to justify the chosen values.
Introduction
Clarity and Focus: While the introduction is well-contextualized, it is recommended to reduce redundancies about the concept of fear and place greater emphasis on the study's practical implications. This would help readers quickly identify the study's relevance.
Practical Implications: Including an additional paragraph on how the results could inform preparation for future pandemics or health crises would enhance the introduction.
Results Visual Clarity: Simplifying or combining certain tables would improve comprehension. For example, adhering to the journal's guidelines and adopting a consistent table style would refine the presentation (e.g., improving Table 1 and summarizing redundant information across the English and French FCV-19S versions).
Threshold Interpretation: Expanding the discussion on Youden's index values and the selection of cut-off points (ROC) for sensitivity and specificity metrics would better contextualize the results.
Discussion, Connection to Previous Studies: Although the results are contextualized in relation to previous studies, it would be useful to highlight how this work provides novel contributions to the field.
Limitations and Future Directions: Elaborating on specific limitations of the FCV-19S in the nursing context would be valuable. Suggesting future research to adapt this instrument to other settings or to evaluate additional properties, such as test-retest reliability, would enrich the discussion.
Practical Applications: The discussion could benefit from a more explicit focus on how the results can translate into actionable steps to improve healthcare workers' well-being.
Conclusions, The conclusions are clear and relevant but could be strengthened by emphasizing how the findings could guide the development of specific interventions to reduce fear and stress among nursing staff during future health crises.
I hope these suggestions are helpful to the authors in the process of revising and improving the manuscript. The article addresses an important topic and has great potential to contribute to the understanding of psychometric factors associated with fear of COVID-19 among nursing staff.
Sincerely
Author Response
Comment 1: Dear Editor, Thank you for the opportunity to review the manuscript titled "Validation of the Canadian English and French versions of the Fear of COVID-19 Scale in Quebec Nursing Staff." Below, I present several suggestions that I believe could enhance the manuscript and improve its academic impact: Study Design. The use of a cross-sectional design is appropriate for psychometric studies aimed at validating instruments in a specific context. Including a diverse sample of nursing staff from Quebec enhances the representativeness and generalizability of the results within this group. Sampling and Sample Size, The probabilistic method for participant selection strengthens external validity. Although the expected participation rate (10%) is low, it is typical for online surveys and was handled appropriately. Instruments, The selection of validated scales, such as the FCV-19S, PCL-5, and GAD-7, ensures the quality of the measurements. The use of both English and French versions reflects adequate cultural and linguistic sensitivity to the Canadian context. Statistical Analysis, The combination of internal consistency analysis (Cronbach's alpha), structural validation (exploratory factor analysis), and ROC analysis demonstrates a comprehensive approach to evaluating psychometric properties. The interpretation of sensitivity, specificity, and AUC (area under the curve) values is appropriate.
Response 1: We want to thank the reviewer for their encouraging feedback and hope our response was able to address their concerns.
Comment 2: Areas for Improvement While a probabilistic method is mentioned, it is not entirely clear how potential selection or non-response biases were addressed. Including an analysis comparing participants and the target population would strengthen the manuscript.
Response 2: In the section discussing the strengths and limitations of the study, we have now addressed the potential for selection or non-response biases by including an analysis comparing the participants to the target population. This comparison is based on sociodemographic characteristics made publicly available by their professional order. (L417-422)
Comment 3: Instruments, The article mentions that the FCV-19S was developed for the general population rather than nursing staff. It would be helpful to further discuss the implications of this limitation and how the instrument could be adapted for healthcare professionals.
Response 3: Following the reviewer's comment, we have expanded the discussion section to include potential adaptations of the Fear of COVID-19 Scale, and have highlighted the unique fears that healthcare workers, such as nursing staff, may face. (L402-414)
Comment 4: Study Duration, Although the data collection period (June to October 2021) is reasonable, there is no detail on how contextual changes (e.g., health policy updates or vaccine availability) were managed, which may have influenced participant responses.
Response 4: In response to the reviewer's comment, we've included a new paragraph towards the end of the introduction that highlights the impact of COVID-19 on our population during the study period. (L103-113)
Comment 5: Contextual Variables, Expanding on how workplace characteristics (e.g., perceived stress levels or workload during the pandemic) were controlled or analyzed as covariates would add depth to the analysis.
Response 5: We want to thank the reviewer for their insightful comment. In our study, we focused on the psychometric properties of the FCV-19S, specifically assessing its internal consistency, structural validity, convergent validity, and criterion validity. While workplace characteristics such as perceived stress levels and workload are indeed important factors, they were not within the scope of this psychometric analysis and were not assessed. Following the reviewer's comment, we've added a sentence to the limitations section of our study. We note that future studies assessing the fear of COVID-19 among nursing staff should consider including covariates such as workplace characteristics to better understand how these factors may interact with fear levels (L427-431)
Comment 6: Threshold Justification The selection of thresholds for interpreting the FCV-19S's sensitivity and specificity could benefit from a more detailed theoretical or empirical discussion to justify the chosen values.
Response 6: Following the reviewer’s comment, we have now added a paragraph in the Discussion section in which we discuss our choice of threshold. (L341-352)
Comment 7: Introduction. Clarity and Focus: While the introduction is well-contextualized, it is recommended to reduce redundancies about the concept of fear and place greater emphasis on the study's practical implications. This would help readers quickly identify the study's relevance.
Response 7: To improve clarity and focus, we have combined the first two sentence explaining fear into a single statement (L49-51). We also clarified the study’s relevance by highlighting how early detection of heightened fear responses and the establishment of a clear threshold can aid in timely interventions and improve the allocation of mental health support resources. (L98-101)
Comment 8: Practical Implications: Including an additional paragraph on how the results could inform preparation for future pandemics or health crises would enhance the introduction.
Response 8 : Following the reviewer’s comment, we have further expanded in the Introduction on how the results could inform preparation for future pandemics or health crises. (L95-101)
Comment 9: Results Visual Clarity: Simplifying or combining certain tables would improve comprehension. For example, adhering to the journal's guidelines and adopting a consistent table style would refine the presentation (e.g., improving Table 1 and summarizing redundant information across the English and French FCV-19S versions).
Response 9: We simplified Table 1 by removing the French version of the items and adhering to the journal's guideline that figures and tables must be in English only. (L251-256)
Comment 10: Threshold Interpretation: Expanding the discussion on Youden's index values and the selection of cut-off points (ROC) for sensitivity and specificity metrics would better contextualize the results.
Response 10: Following the reviewer’s comment, we have now added a paragraph in the Discussion section in which we discuss our choice of threshold in relation to Youden’s index. (L341-352)
Comment 11: Discussion, Connection to Previous Studies: Although the results are contextualized in relation to previous studies, it would be useful to highlight how this work provides novel contributions to the field.
Response 11: Following the reviewer’s comment, we highlight in the first paragraph of the Discussion the novel contributions of the study to the field. (L321-324)
Comment 12: Limitations and Future Directions: Elaborating on specific limitations of the FCV-19S in the nursing context would be valuable. Suggesting future research to adapt this instrument to other settings or to evaluate additional properties, such as test-retest reliability, would enrich the discussion.
Response 12: Following the reviewer’s comment, we have further elaborated in the Discussion section on the limitations of using FCV-19S among nursing staff (when discussing issues related to the relatively low scores obtained), and on the need to evaluate additional psychometric properties. (L402-414; L435-438)
Comment 13: Practical Applications: The discussion could benefit from a more explicit focus on how the results can translate into actionable steps to improve healthcare workers' well-being.
Response 13 : Following the reviewer’s comment, we have now added a paragraph in the Discussion section regarding how our results related to perceptions of preparedness and control at work may support actionable steps to improve nursing staff’s well-being. (L360-374)
Comment 14: Conclusions, The conclusions are clear and relevant but could be strengthened by emphasizing how the findings could guide the development of specific interventions to reduce fear and stress among nursing staff during future health crises.
Response 14: We have now added a sentence emphasizing how the findings could guide future intervention development. (L444-446)
Comment 15: I hope these suggestions are helpful to the authors in the process of revising and improving the manuscript. The article addresses an important topic and has great potential to contribute to the understanding of psychometric factors associated with fear of COVID-19 among nursing staff. Sincerely
Response: 15: We thank once again the reviewer for their comments.
Round 2
Reviewer 3 Report
Comments and Suggestions for Authors
Dear Author(s),
Congratulations on your interesting work! I have read the revised version and am delighted to see the comprehensive revision. My minor concern pertains to the differences in the previous psychometric properties of the Fear of COVID-19 Scale within the English-speaking population and in your study. Please elaborate on and highlight the novelty of your study.
Author Response
Comments 1: Congratulations on your interesting work! I have read the revised version and am delighted to see the comprehensive revision. My minor concern pertains to the differences in the previous psychometric properties of the Fear of COVID-19 Scale within the English-speaking population and in your study. Please elaborate on and highlight the novelty of your study.
Response 1: Thank you for your positive feedback and for recognizing our efforts in revising the manuscript. We appreciate your suggestion to elaborate on the differences in the previous psychometric properties of the Fear of COVID-19 Scale within English-speaking populations. In response, we have clarified the comparison between our previous validation study and the current findings, explicitly highlighting the stability of the scale’s psychometric properties over time and across different samples. Additionally, we have emphasized the novelty of this study by discussing its unique contributions, particularly the validation within a nursing staff sample (L353-364) and the use of ROC analyses (L321-324).